# Determinants of Non-Timber Forest Product Planting, Development, and Trading: Case Study in Central Vietnam

**Thanh Van Nguyen** [1,2], **Jie Hua Lv** [1,*], **Thi Thanh Huyen Vu** [1,2] **and Bin Zhang** [1]

1   School of Economics and Management, Northeast Forestry University, Harbin 150040, China;
    ntvan3588@gmail.com (T.V.N.); thanhhuyenkt.vuthi83@gmail.com (T.T.H.V.); fyz1025@126.com (B.Z.)
2   Saodo University, Hai Duong 03000, Vietnam
*   Correspondence: lvjiehuajiaoshou@126.com; Tel.: +86-1394-6065-303

**Abstract:** Non-timber forest products (NTFPs) play an active role in economic development, improving household livelihoods, raising the value of forestry production, and supporting sustainable forest management. This study involved a comprehensive assessment of the growth, development, and trade of NTFPs in Vietnam by combining logistic and tobit methods. Surveys were used to interview 400 households in three regions of Central Vietnam. Results showed that the planting, development, and trading of NTFPs are shaped by forestry production experience, the number of laborers, the percentage of wage earners, agricultural income, timber income, per capita income, the presence of bank deposits, the distance between forest and house, an understanding of forestry economic policies, and participation in technical training. Each factor had a different level of influence. Among the six NTFP groups, the groups generating yarn and medicines produced the highest income and had a strong impact on household reliance on NTFPs. This was followed by NTFPs used to generate food, oil, and plastic. The proportion of people with wages, and the income variable system, negatively impacted NTFP planting and income generation, which reduced household reliance on NTFPs. This means that there is a trade-off between NTFPs and other income generating activities. In the future, the government should develop specific plans, policies, and strategies for developing each type of NTFP suitable to each region's natural conditions. The policies should include supporting people with low-interest bank loans; expanding the number of training courses to increase their understanding of forestry economic policies; and implementing cultivation techniques and forest care to improve the productivity, quality, and efficiency of NTFP products.

**Keywords:** Vietnam; non-timber forest products; NTFPs; logistic; tobit method

## 1. Introduction

Non-timber forest products (NTFPs) are an important part of forest ecosystems, particularly in tropical natural forests [1]. In Vietnam, NTFPs have become a central source of cash income for people living in local regions near forests [2]. Vietnam currently has approximately 14.5 million hectares of forest, of which 10.3 million hectares consist of natural forest. Since 2014, the government has stopped timber exploitation from natural forests nationwide [3]. This plan has significantly reduced the volume of wood production in Vietnam. NTFPs are a major alternative to or complement timber harvesting and other resource industries. Vietnam NTFPs are exported to about 90 countries. In 2018, NTFPs' exports continued to achieve positive results, with a turnover of 480 million USD. This was an increase of 2.47 times compared to that of 2013 (194 million USD) [4]. NTFP exploitation and processing have attracted thousands of laborers, mainly in rural and mountainous areas, significantly reducing poverty in forested localities.

Over the past two decades, the Vietnamese government has implemented several afforestation and development programs [5]. Several programs, such as Programs 327, 611, and FSDP (WB3), have tried to encourage replanting, protect forests, improve land utilization, raise living standards, and support the settlement program of the government [6–9]. Given this, Vietnam's forest area expanded by approximately 5 Mha in 25 years from 1990 to 2015 (from 9 Mha in 1990 to 14 Mha in 2015). It is anticipated that this land coverage will increase to 4.15 Mha by 2020 [10]. Planting activities have played a significant role in increasing the development of NTFPs. Despite this increase, the contribution of NTFPs remains lower than the potential, accounting for less than 10% of total forestry export turnover [4]. NTFP cultivation, exploitation, and processing practices are outdated and mainly based on indigenous experience and knowledge. Further, productivity, efficiency, and product quality are low due to outdated equipment. NTFPs have a very scattered, fragmented, and spontaneous distribution, with no specific planning strategy [11]. Therefore, it is important to identify the factors that will improve the efficiency of planting and trading NTFPs. The goal is to achieve economic transformation, while substantially reducing lumber production volume and improving household livelihoods.

The role of NTFPs in the livelihoods of rural households in developing countries has received increasing attention from scientific communities and policy makers. NTFPs provide many rural and urban communities with subsistence needs, such as food, health (medicines), building materials, and energy [12–15]. In addition, they are also essential for generating cash income [13,16,17]. The contribution of NTFP transactions to household cash income varied from 20% to 50% [12,18,19]. The use of NTFPs is the key to achieving sustainable forest management [20]. There have been very few studies on NTFPs in Vietnam. Studies to date have mainly focused on NTFP classification [21–24]. A study by Viet Quang and Nam Anh [2] evaluated the dependence of forest dwellers on NTFPs and identified the relationship between household characteristics and cash income generated by NTFP collection. Reforestation studies have mainly focused on timber tree products [25,26]; no comprehensive research has been conducted on the factors influencing NTFPs.

Some previous studies often considered the adoption of tree planting as a dichotomous or limited dependent variable in probit, logit, Heckman, or tobit regression models [27–31]. In this study, we selected a sample of 400 households to assess the factors affecting NTFP planting and development in Central Vietnam by using a logistic estimation method. When assessing the factors that affect people's income and dependence on NTFPs, we only considered 279 (out of 400 surveyed samples) households that participated in NTFP planting. Therefore, the dependent variable contained a partial value observed with a positive probability of 0 (121 households not involved in NTFPs planting). In this case, the tobit estimation method would be more appropriate. The main purpose of this study was to assess the growth, development, and trade of NTFPs in Vietnam by combining logistic and tobit methods.

The study has two major objectives: (i) using the logistic model to analyze the factors affecting decisions related to planting and developing high economic value NTFPs at the household level in three regions of Central Vietnam and (ii) using the tobit model, which defines the relationship between households and NTFPs, to assess the determinants of household income from NTFPs, analyze household dependence on NTFPs (calculated as a percentage of total income), and identify NTFP groups that generate high income.

This study emphasizes the role of NTFPs in stabilizing farm household income, playing an active role in economic development, and contributing to raising the value of forestry production in the central region of Vietnam. Sustainable forest management is defined as the foundation for the development of Vietnam's forestry sector. NTFPs are resource-based products. This understanding is important in guiding and enabling the design of appropriate strategies and policies to enhance the performance of the country's trade of NTFP products and sustainable forest management in countries like Vietnam, as they restructure the forestry sector and move toward fulfilling Reducing Emissions from Deforestation and Forest Degradation (REDD+) agreements [32].

The rest of this paper is organized into three sections. Section 2 highlights the method, research subjects, variable set-up, and data for the analyses. Section 3 includes the interpretation of the results and presents the discussion. The final section summarizes the conclusions and proposes key policies.

## 2. Methods and Data

### 2.1. Research Objects

This study was conducted in three regions of Central Vietnam, with a focus on forest areas adjacent to the households, particularly areas less than 10 km radius from the forest area to households. Survey data were collected from the results of the project "Investigating and evaluating the current situation and proposing the development of a number of non-timber forest products of high economic value in the central region and central highlands for the project of restructuring the industry forestry". The study area covered 10 mountainous provinces of three diverse geographical regions in Vietnam: North Central (Thanh Hoa, Nghe An, Ha Tinh, and Quang Tri), South Central (Quang Nam, Binh Dinh, and Khanh Hoa), and Central Highlands (Kon Tum, Gia Lai, and Lam Dong). Most households in this area are engaged in agriculture, forestry, and fisheries. A common feature of these mountainous upland sites is the presence of large forest areas [33,34].

NTFP resources in the three central regions are abundant, with an average of 20–60 species per province. The resources include six main groups, including yarn, food, medicinal uses, oil and plastic, essential oils, and others. While the output is relatively low and unfocused, there are many species of high economic value, including bamboo, rattan, *Amomum*, *Litsea*, and cinnamon. These contribute to the economic income of mountainous people, ranging from 5%–10% of the total household income and reaching up to 40% in some locations [1].

### 2.2. Survey Method

We conducted interviews with households living near forests in three areas of Central Vietnam from July 2016 to December 2017. The research questions were investigated within the mixed-method framework that used different tools and approaches for collecting quantitative and qualitative data from field respondents. There were 400 interviews conducted with different households randomly selected in 40 villages in 20 communes. The questionnaire design focused on household behaviors with respect to the planting and trading of high economic value NTFPs. The questionnaire was tested with multiple households in one village before it was applied in other villages. In Vietnam, this procedure ensured that the surveyed households represented the rural population in different agro-ecological regions.

The results showed that out of 400 farmer households, 279 households conducted NTFP planting and trading activities. Of these, the North Central Coast had 128 households, the South Central Coast had 78 households, and the Central Highlands had 73 households (Table A1). Of the 279 households involved in planting and trading NTFPs, most focused on yarns (80 households, accounting for 28.7% of the total households participating in NTFP planting and trading) and medicines (78 households, accounting for 28% of the total households participating in NTFP planting and trading). The number of households participating in the planting and trading of NTFPs for food (52 households, 18.6%) and oil and plastic (57 households, 20.4%) was also relatively high. The number of households participating in the planting and trading of NTFPs for essential oils and other items was not large, with 29 households (10.4%) for the essential oil group and 19 households (6.8%) for the other group.

Furthermore, the North Central region had a large number of NTFP species being harvested by people. In particular, the group with the most diverse species was the medicinal use group, with approximately 17 to 21 species. The group producing fibers was also quite diverse in species composition, with the highest number of species (10 species) in Thanh Hoa and Nghe An provinces. The medicinal group was strong in the South Central region; the food group was abundant in Quang Nam and Khanh Hoa (4–9 species); and the resin group was well-represented in Khanh Hoa, Binh Dinh, and Da Nang (1–5 species). The Central Highlands had a diverse number of NTFP species being harvested and traded. In particular, the medicinal plant species group accounted for the largest number of species, with a total of 35 species harvested. This was followed by the yarn and food groups, and the remaining two groups (oil and plastic, and other). Thus, all three regions of Central Vietnam have a significant number of NTFP species for yarn, medicinal, food, and oil and plastic use. Attention is needed to develop their full potential (Table A2).

*2.3. Model Selection*

- Analyze the factors affecting the decisions related to planting and developing high economic value NTFPs:

This study applied the binary logistic regression model to analyze the factors affecting decisions related to planting and developing high economic value NTFPs in farmer households in the three central regions of Vietnam (Model 1). The logistic regression model in the study is described as follows. Household behaviors in the planting and developing of NTFPs with high economic value are assigned two conditions: "planting" (value = 1) and "non-planting" (value = 0). The explained variable features of non-continuity are a binary choice problem. The binary logit model is presented in Equation (1):

$$\ln\left(\frac{P}{1-P}\right) = \beta_0 + \sum_{i=1}^{n} \beta_i x_i + \mu. \tag{1}$$

In this expression, P is the probability that a worker household selects to plant and develop NTFPs with high economic value, $\beta_0$ represents the regression intercept, $x_i$ represents the ith factor influencing the behavior of a worker household, $\beta_i$ represents the regression coefficient of the ith factor, and $\mu$ is the random disturbance term.

- Analyze the factors affecting NTFP participation by households involved in NTFP planting:

After analyzing and evaluating the factors that influence NTFP planting and development decisions, we applied the tobit model to assess the factors affecting NTFP participation by households involved in NTFP planting. Tobin created the tobit model [35] to solve problems with specific econometric data. For n observations, the tobit model consists of latent variables $y_i^*$ and observed variables $y_i$. Formally, the latent variables $y_i^*$, i = 1, 2, ... , *n*.

$$y_i^* = x_i' \beta + u_i. \tag{2}$$

Observations $y_i$, i = 1, 2, ... , *n* are derived as:

$$y = \begin{cases} y_i^*, & \text{if } y_i^* > 0 \\ 0, & \text{if } y_i^* \leq 0 \end{cases}. \tag{3}$$

Here, $u_i$ are i.i.d. normally distributed random variables, with the expectation of 0 and unknown variance $\sigma^2$; $\beta$ is the unknown vector of parameters to be estimated; and $x_i$ is a known vector of regression variables for the ith observation. The model is a mixed model, consisting of the regression in Equation (2) and the threshold component in Equation (3). This is very similar to a probit model [36].

In this research, the two dependent variables used in the tobit model are:

(1)  NTFP farmer household income, measured as the sales value of NTFPs (Model 2).
(2)  Dependence of households on NTFPs, measured as the percentage of NTFP-generated income to total household income (Model 3).

Both NTFP income and dependence of households on NTFPs are used for regression with the same group of independent variables using the tobit model, because the sample data of these dependent variables are censored at zero [37]. Models 2 and 3 were tested with the sample of households who participated in the Model 1 test.

*2.4. Index Selection*

After investigating the actual situation of planting, developing, and trading NTFPs in Central Vietnam and combining research data collection and evaluation of previous studies on NTFPs [20,27,38–40] and tree planting research [41–44], we decided to use Model 1 to select 21 factors (independent variables), classified into 5 groups: household head characteristics, household workforce characteristics, household economic characteristics, woodland characteristics, and other external characteristics. Assuming that only households engaged in planting and developing NTFPs of high economic value will generate NTFP income. Models 2 and 3 use the same independent variables as Model 1 and add an independent variable about the NTFP type with which the households participated. The purpose of this was to assess which NTFP group brings the highest economic efficiency to the people.

The variable design, measurement index coding, and expected effect of each variable used in the three models are detailed in Table 1. Detailed description statistics are shown in Table A3.

**Table 1.** Variable explanation and expected effect direction.

| Variable Type | Model | Indicator Coding | Variable Explanation | Expected Effect Direction |
|---|---|---|---|---|
| **Explained variables** | | | | |
| Behavior related to planting and developing NTFPs | $M_1$ | PLAN | Whether or not household is engaging in planting and developing NTFPs: 1 = yes, 0 = no | |
| NTFP income | $M_2$ | $INCO_{NTFPs}$ | The sale value of NTFPs (Thousand dong) | |
| Dependence of households on NTFPs | $M_3$ | PERC | The percentage of income generated by NTFPs in the total income of the household (%) | |
| **Explanatory variables** | | | | |
| Household head characteristics (HEAD) | $M_{1,2,3}$ | AGE | Age of household head (years) | Negative (−) |
| | $M_{1,2,3}$ | EDUC | Educational level of household head (years) | Positive (+) |
| | $M_{1,2,3}$ | GEND | Gender of household head: 1 = female, 0 = male | Negative (−) |
| Household workforce characteristics (HOUS) | $M_{1,2,3}$ | EXPE | Experience involved in forestry production (years) | Positive (+) |
| | $M_{1,2,3}$ | LABO | Number of laborers in the household (person) | Positive (+) |
| | $M_{1,2,3}$ | FEMA | Number of female laborers in the household (person) | Positive (+) |
| | $M_{1,2,3}$ | EARN | Proportion of wage earners (%) | Negative (−) |
| Household economic characteristics (ECON) | $M_{1,2,3}$ | CAPI | Annual per capita income (Thousand dong) | Positive (+) |
| | $M_{1,2,3}$ | BANK | Whether or not there were bank deposits: 1 = yes, 0 = no | Positive (+) |
| | $M_{1,2,3}$ | $INCO_{AGRI}$ | Income from agriculture (Thousand dong) | Negative (−) |
| | $M_{1,2,3}$ | $INCO_{TIMBER}$ | Income from timber (Thousand dong) | Negative (−) |
| | $M_{1,2,3}$ | $INCO_{WAGE}$ | Income from wage (Thousand dong) | Negative (−) |
| | $M_{1,2,3}$ | $INCO_{OTHER}$ | Other income (Thousand dong) | Negative (−) |
| Woodland characteristics (LAND) | $M_{1,2,3}$ | AREA | Forest area (ha) | Positive (+) |
| | $M_{1,2,3}$ | HOME | Distance between the woodland and home (km) | Negative (−) |
| | $M_{1,2,3}$ | ROAD | Distance between the woodland and road (km) | Negative (−) |
| | $M_{1,2,3}$ | SLOP | Slope of the woodland: 1 = <5 degrees, 2 = 5 to 15 degrees, 3 = 15 to 25 degrees, 4 = 25 to 35 degrees, 5 = >35 degrees | Negative (−) |
| | $M_{1,2,3}$ | SOIL | Soil quality of the woodland: 1 = very poor, 2 = poor, 3 = moderate, 4 = fair, 5 = very fair | Positive (+) |
| Other external characteristics (OTHE) | $M_{1,2,3}$ | INFO | Information of forestry economic policy: 1 = yes, 0 = no | Positive (+) |
| | $M_{1,2,3}$ | TRAI | Whether or not to participate in technical training: 1 = yes, 0 = no | Positive (+) |
| | $M_{1,2,3}$ | INFL | The mutual influence between farmer households: 1 = very weak, 2 = weak, 3 = middle, 4 = strong, 5 = very strong | Positive (+) |
| Regional dummy variable | $M_{1,2,3}$ | REGI | Forest area dummy: 1 = North Central, 2 = South Central, 3 = Central Highlands | Uncertain |
| NTFP type | $M_{2,3}$ | NTFPs | NTFP type: 1 = group for yarn, 2 = group for food, 3 = group for medicinal use, 4 = group for oil and plastic, 5 = group for essential oils, 6 = group for others | Positive (+) |

## 3. Results

### 3.1. Descriptive Statistics of Explanatory Variables

According to the statistical data given in Appendix A (Table A3), the age of household head was 20 to 70 years, of which the average age was 44.6 years. The educational level of household head was averaged at 1.79, which meant that the average education level of the household head was only higher than the primary level and yet to reach the lower secondary level, and that very few household heads had upper secondary or higher education. The reason is because the study area was concentrated in the mountainous and rural areas of Central Vietnam, where the majority of the people are poor ethnic people with no learning conditions. The number of female-headed households (41%) was less than that of male-headed households. Households engaged in forestry production had been the highest for 30 years and some households had never participated in forestry production. The average number of years participating in forestry production of surveyed households was 8.24 years. This is quite a long time and people can gain experience for participating in planting and trading NTFPs. The average number of employees in the household was 3.09, of which the average number of female workers was 1.84. This proved that the number of female workers in the surveyed households also accounted for a large proportion (over 50% of the average household labor). The percentage of wage earners was very low, with an average of 16% of the total number of employees. The annual household per capita income was 34.71 million Vietnamese dong. The majority of households did not have bank deposits and the number of households with bank deposits accounted for 29% of the sample. Households had an average forest area of 5.21 ha. The average distance from home to the forest was 5.85 km and the average distance from the house to the road was 5 km. The quality of forest land was at an average of 2.16, meaning that most of the land used for planting forestry trees was of poor quality. The forest slope was at an average of 2.93, meaning that the average forest slope ranged from 15 to 25 degrees. The number of households with forestry economic policy information reached 47%. The number of households participating in forestry training reached 45%. The influence of farmers between households reached an average of 2.71, meaning that the farmers had medium influence on each other. There was no close relationship between farming households in NTFP planting and trading. The number of households participating in planting and trading of NTFPs reached 69.7%. The surveyed households in the North Central Coast region accounted for 40% and in the South Central region and Central Highlands accounted for 30%.

### 3.2. Regression Model Results

The aggregated data from the survey were entered into EViews 8 software to run the regression model using the ML binary logit method and ML censored normal method (tobit). Results are shown in Table 2. Specifically, columns 2 and 3 show the results from estimating the factors affecting the planting and development of NTFPs. This model had a determination coefficient of $R^2 = 54.45\%$, $p = 0.00 < 0.1$. The full model had a forecast rate of 89%, which corresponded to 356/400 households being correctly forecasted. This indicated that the testing model was acceptable. The Wald test was used in all the three models to examine the role of the variables and to examine autocorrelation and variance to determine model accuracy and validity. The values of Wald-chi2 in all the three models ($p = 0.00 < 0.05$) indicated that these models were significant in explaining the variations of the dependent variable. Columns 4 and 5 present the results from estimating the factors affecting NTFP income. Columns 6 and 7 show the factors affecting household dependence on NTFPs.

**Table 2.** Model estimation results.

| Variable (1) | Model 1: Behavior of Planting and Developing NTFPs | | Model 2: Determinants of Income from NTFPs | | Model 3: Dependence of Households on NTFPs | |
| --- | --- | --- | --- | --- | --- | --- |
| | Coefficient (2) | z-Statistic (3) | Coefficient (4) | z-Statistic (5) | Coefficient (6) | z-Statistic (7) |
| C | −6.05 *** | −2.75 | −17.22 *** | −3.51 | −0.90 | −0.16 |
| AGE | 0.01 | 0.38 | −0.01 | −0.17 | −0.02 | −0.41 |
| EDUC | 0.24 | 0.69 | 0.30 | 0.36 | −0.29 | −0.30 |
| GEND | −0.11 | −0.31 | 0.19 | 0.21 | −0.14 | −0.15 |
| EXPE | 0.28 *** | 5.04 | 0.63 *** | 6.90 | 0.61 *** | 5.88 |
| LABO | 0.99 *** | 2.86 | 5.00 *** | 7.11 | 2.65 *** | 3.33 |
| FEMA | 0.02 | 0.07 | −1.43 * | −1.76 | −1.11 | −1.21 |
| EARN | −7.30 *** | −6.05 | −8.42 ** | −2.39 | −10.77 *** | −2.72 |
| INCO$_{AGRI}$ | −0.02 ** | −2.09 | −0.12 *** | −6.24 | −0.14 *** | −6.72 |
| INCO$_{TIMBER}$ | −0.03 ** | −2.34 | −0.15 *** | −5.75 | −0.16 *** | −5.70 |
| INCO$_{WAGE}$ | 0.02 | 1.42 | −0.10 ** | −2.37 | −0.09 * | −1.81 |
| INCO$_{OTHER}$ | −0.14 | −0.86 | −1.17 ** | −2.22 | −0.25 | −0.43 |
| CAPI | 0.06 ** | 2.18 | 0.39 *** | 6.90 | 0.22 *** | 3.49 |
| BANK | 1.34 *** | 2.96 | 3.54 *** | 3.59 | 3.81 *** | 3.42 |
| AREA | 0.08 | 1.61 | −0.02 | −0.19 | −0.01 | −0.05 |
| HOME | −0.51 *** | −5.95 | −0.88 *** | −4.61 | −0.86 *** | −3.99 |
| ROAD | 0.14 | 1.54 | 0.11 | 0.50 | −0.08 | −0.30 |
| SOIL | 1.42 *** | 5.63 | 0.52 | 1.12 | 0.12 | 0.23 |
| SLOP | −0.13 | −0.37 | −1.52 * | −1.77 | −1.66 * | −1.72 |
| INFO | 1.69 *** | 4.26 | 3.54 *** | 3.98 | 3.67 *** | 3.65 |
| TRAI | 1.68 *** | 3.80 | 5.87 *** | 6.25 | 4.86 *** | 4.60 |
| INFL | −0.09 | −0.46 | −0.34 | −0.75 | 0.51 | 0.10 |
| REGI1 | 1.07 ** | 2.10 | −0.68 | −0.56 | −1.41 | −1.04 |
| REGI2 | 0.94 * | 1.77 | 1.74 | 1.25 | −0.29 | −0.19 |
| NTFP1 | | | 15.03 *** | 10.96 | 16.35 *** | 10.44 |
| NTFP2 | | | 8.99 *** | 6.68 | 11.35 *** | 7.44 |
| NTFP3 | | | 11.62 *** | 8.98 | 13.95 *** | 9.55 |
| NTFP4 | | | 8.19 *** | 5.74 | 10.76 *** | 6.72 |
| NTFP5 | | | 6.24 *** | 3.53 | 6.99 *** | 3.51 |
| NTFP6 | | | 7.33 *** | 3.63 | 9.14 *** | 3.10 |
| McFadden R-squared | 0.5445 | | | | | |
| Prob (LR statistic) | 0.00 | | | | | |
| Wald-chi2: Prob > chi2 | 0.00 | | 0.00 | | 0.00 | |
| Observations | 400 (censored observations = 121, uncensored observations = 279) | | | | | |

\* Significant at 10%; \*\* Significant at 5%; \*\*\* Significant at 1%.

The results of Table 2 indicated that the age of household head, the educational level of household head, and the gender of household head did not affect the behavior of planting and developing NTFPs with high economic value of farmer households and NTFP income, and did not decide the dependence of households on NTFPs. The higher the number of years a household had engaged in forestry production and the greater the number of employees, the more active it was in planting trees and generating more NTFP income and a higher percentage of NTFP income. This was shown by the positive coefficients of EXPE and LABO variables in all three models. The number of female laborers in the household did not affect the behavior of planting and developing NTFPs with high economic value of farmer households and did not decide the dependence of households on NTFPs, but had a negative impact on households' NTFP income at a significant level of 10% ($\beta_{FEMA} = -1.43$). The proportion of wage earners had a negative influence on the decision of planting and developing NTFPs with high economic value of farmer households, which was shown by the negative coefficient ($\beta_{EARN} = -7.30$), but wage income had no effect on this decision. Both of these variables negatively affected NTFP income and reduced household dependence on NTFPs. This was shown by the negative coefficients of EARN and INCO$_{WAGE}$ variables in Models 2 and 3. The income from agriculture and timber adversely affected the behavior of planting and developing NTFPs with high economic value of farmer households, thereby reducing NTFP income and the dependence of households on NTFPs. This was shown by the negative coefficients of INCO$_{AGRI}$ and INCO$_{TIMBER}$ variables in all three models. Other income only negatively affected NTFP income ($\beta_{OTHER} = -1.17$) at the 95% significance level,

but did not affect the behavior of planting and developing NTFPs with high economic value of farmer households and did not decide the dependence of households on NTFPs. The greater the per capita income and the number of households with bank deposits, greater will be the number of farmers motivated to participate in afforestation, increase in income from NTFPs, and increase in the percentage of NTFP income in total household income. This was represented by the positive coefficients of CAPI and BANK variables in all three models. Forest area was not significant in all three models. The distance between the woodland and home had a negative impact on planting and developing NTFPs of high economic value, and reduced NTFP income and household dependence on NTFPs. This was shown by the negative coefficients of HOME and ROAD variables in all three models. There was no basis to confirm that the distance between the woodland and the road had an effect on this. The slope of the woodland had a positive influence on planting and developing NTFPs with high economic value of farmer households ($\beta_{SOIL}$ = 1.42) at the 99% significance level. The quality of the woodland had a positive impact on increasing NTFP revenue and household dependence on NTFPs. This was represented by the positive coefficient of SLOP variable in Models 2 and 3. When farmers have an understanding of forestry policies and are basically trained in forestry techniques, farmers will become more actively involved in planting and developing NTFPs of high economic value; at the same time, it will increase NTFP revenue and the percentage of NTFP income in total household income. This was represented by the positive coefficients of INFO and TRAI variables in all three models. The mutual influence between farmers was not large enough to influence NTFP planting of households and it did not affect the NTFP income of the people, as there was no impact on assessing the reliance on NTFPs of households. The forest area dummy affected the decision of planting and developing NTFPs with high economic value of farmer households, but there was no effect on NTFP income and household dependence on NTFPs. The dummy variable for the NTFP types used in Models 2 and 3 also had a positive impact on increasing household NTFP income and household dependence on NTFPs. This was represented by the positive coefficients of the NTFP variables in models.

## 4. Discussion

This is the first in-depth study in Vietnam to combine the logistic and tobit estimation methods to analyze the factors affecting NTFPs. We used 23 factors (including 2 regional dummy variables) to analyze the factors affecting NTFP planting and added six NTFP type variables to assess in detail the impact of each NTFP group on income and the percentage of household income derived from NTFPs. This is one way in which our research differed from previous studies. This study's results indicated that there are 13 factors (including 2 regional dummy variables) influencing NTFP planting and development decisions, 20 factors affecting the income gains from NTFPs, and 18 determinants of household dependence on NTFPs (including 6 dummy variables for NTFP type). The results of this study complement previous studies [2,21–24] by assessing the factors affecting NTFPs in Vietnam on two aspects: planting and income from NTFP. This result has important implications for the development of an NTFP-focused economy, forest conservation, and sustainable development of forestry economy in three central regions of Vietnam.

### 4.1. Household Head Characteristics

When considering the impact of household head characteristics on NTFP planting, none of the three variables representing household head impacted the three models ($p_{AGE}$, $p_{EDUC}$, $p_{GEND}$ > 0.1). This result is consistent with other studies [27,30,31,42,44–46]. Educational level also had no significant effect on the decision to join NTFPs. This may be because education is insufficient to affect labor-intensive livelihoods, as they only received a primary education. The relationship between household female roles and household decisions on NTFP utilization has been a significant research focus on NTFP utilization [2,47,48]; however, debate continues on the topic. This study also considered the role of the number of female laborers in the family in planting, developing, and trading NTFP decisions. This study's regression results revealed that the result of this indicator was insignificant, indicating low

reliability. The average age of the household head (44 years) was also not significantly affected by the decision to participate in planting and developing NTFPs.

*4.2. Household Workforce Characteristics*

With regard to the impact of household workforce characteristics, two variables had a statistically significant and positive impact on all three models: the experience involved in forestry production (EXPE) and the number of laborers in the household (LABO). This is logical, because tree planting is labor-intensive. In fact, Vietnam has many advantages for NTFP development. In particular, the agriculture and forestry labor force is very large. In 2018, the total labor force in agriculture, forestry, and fishery in Vietnam was 21,565 people, accounting for 40.2% of the country's total labor force. The people themselves have a lot of experience and indigenous knowledge about planting, harvesting, preserving, processing, and using NTFPs [49].

The proportion of wage earners (EARN) had a strong negative effect in the three models ($p_{EARN}$ < 0). If the number of people with fixed income in a household is high, it is more likely that the household will not plant trees. This reduces income and household dependence on NTFPs. This may be because people are working full-time for the government and businesses, leaving no time for planting, developing, and trading NTFPs [50].

The number of female workers (FEMA) did not affect Models 1 and 3, but had a negative effect in Model 2. This means that if there are more female workers in a household, the income generated from NTFPs will be less. This study differed from those of Viet Quang and Nam Anh [2], who found that households with more female workers will generate more cash income from NTFPs. In this regard, our research focused not only on the types of NTFPs available and harvested in natural forests, but also on NTFPs of high economic value that were selected and planted, generating income. Planting and tending trees are laborious and households with more male employees may be better positioned for this job. As such, the income from NTFPs will be higher. Therefore, our research found that the number of female workers negatively impacted NTFP income. There was not enough evidence to prove that the number of female workers affected household dependence on NTFPs.

*4.3. Household Economic Characteristics*

With regard to the impact of household economic characteristics, income from agriculture ($INCO_{AGRI}$) and income from timber ($INCO_{TIMBER}$) negatively impacted all three models; income from wage ($INCO_{WAGE}$) had a negative impact on Models 2 and 3; and other income ($INCO_{OTHER}$) had a negative impact on Model 2. This suggested that there is a trade-off between agriculture activities, planting, and logging timber activities with respect to NTFP planting. As household workers become more engaged in other productive activities, higher income is generated through those activities; at the same time, the number of NTFP-focused workers drop, reducing NTFP revenue. This is reasonable due to labor allocations [51].

The annual per capita income (CAPI) and whether or not there were bank deposits (BANK) positively impacted the three models. This result showed that when the workers' bank deposits are larger, the annual income per capita levels are higher. This means that more people are willing to participate in NTFP planting and development. In reality, participation in planting, developing, and trading NTFPs is a form of business participation. It is a type of investment behavior that requires a certain amount of capital investment. If a farmer household does not have sufficient funds, then it will be unwilling to engage in NTFP-related businesses [27]. Therefore, government support to grant low-interest loans to farmers is essential to provide capital to grow and develop NTFPs.

*4.4. Woodland Characteristics*

With respect to woodland (AREA) characteristics, the forest area did not have a significant impact in any of the three models ($p_{AREA}$ > 0.1). Other studies have shown that woodland area positively impacts afforestation [52]. This study, however, found that the woodland area did not impact household

NTFP planting and development. This is because the Vietnamese government has pursued policy reforms for the past three decades (from 1991 to present). These include the Forest Land Allocation (FLA) policy, which transfers forest management from state-owned forest enterprises to households, communities, and private enterprises [53]. Farmers have resisted FLA programs at the local level, because they perceive it as restricting land access, and are not willing to use land for reforestation purposes [54,55].

The distance between the woodland and home negatively impacted high-value NTFP planting and development, and reduced income from NTFPs and household dependence on NTFPs. This was shown by the negative coefficients of HOME variables in all three models. This indicated that the farther away the woodland is from home, the less willing the worker household is to engage in planting and developing NTFPs. Li [52] also found this to be logical, because the higher the distance, the higher the opportunity costs associated with tree planting.

Woodland soil quality positively influenced planting and developing NTFPs with high economic value in the farmer households. The woodland slope positively impacted NTFP-generated revenue and increased household dependence on NTFPs. This was represented by the positive coefficient of SLOP variable in Models 2 and 3.

Most ex-post empirical economic models in previous studies have not addressed essential biophysical factors [2,20,56]. Therefore, this study included a set of biophysical factors, including woodland soil quality and slope. This is a new contribution compared to previous studies. Results from Model 2 showed that forest slope negatively impacted NTFP-generated income (at 90% confidence). This means that a larger forest slope will generate lower NTFP-related income. This may be because the large forest slope will make it difficult to care for and harvest NTFPs, reduce NTFP revenue, and generate low collection. A study by Dinh et al. [51] about factors affecting tree planting by rural households in Central Highlands of Vietnam found that households tend not to plant trees on grey soil, which is the second best soil type in the region after basaltic soil. This means that the quality of forest land negatively impacts the planting of trees by households. The Lam Dong province, part of the study area, is a famous region for industrial crops, such as tea and coffee. As such, basaltic and grey soils are used for these high-value cash crops. This study was conducted on a larger scale in 10 provinces in Central Vietnam, leading to a difference in the research results. The woodland soil quality (SOIL) had a significant and positive effect on worker household engagement in NTFP planting and harvesting, indicating that the better the woodland soil quality, the more willing people were to engage in NTFP planting and harvesting. This is because better soil quality generates higher yields and better quality in NTFPs [27].

### 4.5. Other External Characteristics

Information about forestry economic policy (INFO) and whether or not to participate in technical training (TRAI) had a significant and positive effect on all three models. This result is consistent with the results of previous studies about NTFPs in several countries, including Asia, for example, in Eastern India by Glendinning [57] and in Bangladesh by Salam [58]; Africa, for example, in Ethiopia by Gebreegziabher [30] and in Ghana by Zhang [46]; and Latin America, for example, in Costa Rica by Schelhas [59]. Farmers participate actively in tree planting programs when their technical knowledge about an agroforestry practice has been enhanced through extension and educational programs. This simultaneously increases NTFP revenue and the percentage of NTFP income to total household income [60]. In Vietnam, NTFP cultivation, exploitation, and processing are outdated and mainly based on indigenous experience and knowledge. Outdated equipment results in low productivity, efficiency, and product quality levels [1].

### 4.6. Regional Dummy Variable

The forest area dummy variables (REGI1 and REGI2) affected the decision to plant and develop NTFPs with high economic value to farmer households ($\beta_{REGI1} = 1.07$ at 95% confidence, $\beta_{REGI2} = 0.94$

at 90% confidence); however, there was no effect on NTFP-generated income and household dependence on NTFPs. Research shows that farming households growing and developing NTFPs of high economic value are most concentrated in the North Central region, followed by the South Central Coast and the Central Highlands region. The North Central Coast includes 96 NTFP species, the South Central region includes 92 species, and the Central Highlands includes 73 species. This difference may be due to different economic and climate conditions in each region, population distribution in different regions, and the awareness of people in different regions in NTFP planting and development. NTFPs in Vietnam are distributed in a very scattered, fragmented, and spontaneous way, with no specific planning [11].

*4.7. NTFP Type*

The dummy variable for the NTFP types used in Models 2 and 3 had a positive impact on household income from NTFPs and an increase in household dependence on NTFPs. This was represented by the positive coefficients of NTFP variables in the models. Our results contribute to insights from previous studies, by including additional variables to the major NTFP groups into the model. The model results showed that the type of NTFP group positively impacted NTFP revenue and increased household dependence on NTFPs (at 99% confidence). The higher the number of households that participate in NTFP production, the greater the income from NTFPs and dependence on NTFPs. The yarn and medicinal use groups were associated with the highest income. The food group and the oil and plastic group generated the next highest income. The essential oils group and other group generated the least amount of NTFP income. The results of the model are consistent with the current practice of NTFP development in Vietnam. Yarn, food, and pharmaceutical products have brought significant revenue to the national economy, especially for mountainous people. The groups that include plants with fibers, medicinal plants, essential oils, and plastic and oils have led to the annual harvest of 350 million bamboo plants, 4500 tons of rattan, 1500 tons of bamboo shoots, 300 tons of fruits, 5000 tons of other products for food, 4500 tons of medicinal herbs, and 130,000 tons of essential oils and plastic [1].

Currently, NTFPs are very developed in the world, many species have a high economic value. Studies by Rajendra Khanal and Krishna B. Bhujei in Nepal [61], Marianne Meijboom in Bhutan [62], and Hongge Zhu in China [27] have shown that NTFPs are especially important for local people's livelihood. In these studies, the authors have not given the species of trees with the highest economic value to the local people. The identification of species of high economic value is important to help the government to plan the development of each forest product in each region. Therefore, the current study introduced the system of variables of NTFP species that are necessary and suitable to the current trend.

Like all studies, this work has some limitations. Many factors influence research on each aspect of NTFP planting, developing, and trading. This study focused on the most decisive factors, consistent with the actual investigative environment in Vietnam. Future studies could use our proposal framework to add other variables to the model. Currently, as the world tries to fight climate change, sustainable forest management plays an important role. Sustainable forest management goes hand in hand with the conservation of biodiversity and meeting basic needs for forest product consumption and export. Therefore, future research should focus on addressing the simultaneous relationship between trade in timber and NTFPs and forest conservation.

## 5. Conclusions and Recommendations

Planting and developing NTFPs are recent new issues in the context of Vietnam's implementation of sustainable forest development policies, while also improving livelihoods for poor people in mountainous areas. This study applied logistic and tobit methods to analyze the factors affecting the decision of households to plant and develop high economic value NTFPs by using field interviews with 400 families in three regions of Central Vietnam. The study determined the relationship between

households and NTFPs by assessing the determinants of household income from NTFPs, analyzed household dependence on NTFPs, and identified NTFP groups that generate high income.

Research results showed that the factors impacting NTFP planting, development, and trading include: forestry production experience, number of laborers, the percentage of wage earners, agricultural income, timber income, per capita income, the presence of a bank deposit, the distance between forest and house, an understanding of forestry economic policies, and participation in technical training. Each factor had a different degree of influence and the results were consistent with the given hypotheses. Special cases were as follows: The number of female employees only affected income from NTFPs. Wage income and forest slope only affected NTFP income and reduced people's dependence on NTFPs, without affecting decisions about NTFP planting. The quality of forest land and forest areas only affected peoples' decisions to plant NTFPs. The specific NTFP group influenced NTFP income and increased household dependence on NTFPs. Among the six NTFP groups, the yarn and medicinal use groups produced the highest NTFP income and significantly impacted household reliance on NTFPs. The next highest income-generating groups were those generating food, and oil and plastic. The proportion of people with wages and the income variable system adversely impacted NTFP planting and income generation, reducing household reliance on NTFPs. This means that there is a trade-off between NTFPs and other income-generating activities. There is a trade-off between the choice of growing and trading timber industry products. Two factors expected to affect NTFP planting and income include the forest land area and the interdependence of households. This study found no such influence. As such, we encourage future studies to test these findings using larger sample sizes and other regions.

The analysis above reveals two recommendations. First, the government should have specific plans and strategies for developing each type of NTFP suitable to the natural conditions of each region, specifically, *Bambusa* sp., *Dendrocalamus barbatus*, *Amomum longiligulare*, and *Cinnamomum cassia* in the North Central region; *Calamus tetradactylus*, *Daemonorops poilanei*, *Morinda officinalis*, and *Machilus odoratissima* in the South Central region; and *Sterculia foetida*, *Macadamia integrifolia*, and *Coix lachryma-jobi* in the Central Highlands. Some NTFP groups with high economic value need to be supported and developed. This includes the yarn and medicinal use groups, which contribute to economic development and poverty reduction, building new rural areas, and ensuring social security. Second, policies can support people with low-interest bank loans and expand the number of training courses to increase their understanding of forestry economic policies, cultivation techniques, and forest care to improve the productivity, quality, and efficiency of NTFP products. This will promote household participation in NTFP planting and development, increase the efficiency of forest product trade, and contribute to sustainable forest development. The goal is to expand the NTFP market and improve export turnover, resulting in the non-timber forest product industry becoming equivalent to the export value of the timber industry.

**Author Contributions:** J.H.L. and T.V.N. helped in conceptualizing the idea of the study design. T.V.N. contributed to the collection of data. T.V.N., T.T.H.V., and B.Z. performed the statistical analysis and contributed to writing (review and editing). All authors have read and agreed to the published version of the manuscript.

**Funding:** The study presented in this paper is supported by the National Social Science Foundation, China (grant number 13BJY032).

**Acknowledgments:** We would like to acknowledge the financial support from the National Social Science Foundation, China, as well as assistance from the School of Economics and Management of the Northeast Forestry University. We are grateful to Van Dinh Nguyen in the Vietnam Academy of Forest Sciences and Cong Chi Tran in the Vietnam Forestry University for support during the field trips. The authors are grateful to the Area Editor and anonymous reviewers whose comments have contributed to improving the quality of this paper.

**Conflicts of Interest:** The authors declare no conflicts of interest.

# Appendix A

Table A1. Participation of farmers' household in NTFP planting.

| Activity Category | | Total | | North Central | | South Central | | Central Highlands | |
|---|---|---|---|---|---|---|---|---|---|
| | | Number of Households (Household) | Proportion (%) | Number of Households (Household) | Proportion (%) | Number of Households (Household) | Proportion (%) | Number of Households (Household) | Proportion (%) |
| No planting of NTFPs | | 121 | 30.25 | 32 | 20.0 | 42 | 35.0 | 47 | 39.2 |
| | Planting NTFPs | 279 | 79.75 | 128 | 80.0 | 78 | 65.0 | 73 | 60.8 |
| Inside: | Group for yarn | 80 | 28.7 | 40 | 31.3 | 24 | 30.8 | 10 | 13.7 |
| | Group for food | 52 | 18.6 | 19 | 14.8 | 18 | 23.1 | 15 | 20.5 |
| | Group for medicinal use | 78 | 28.0 | 27 | 21.1 | 30 | 38.5 | 31 | 42.5 |
| | Group for oil and plastic | 57 | 20.4 | 23 | 18.0 | 14 | 17.9 | 16 | 21.9 |
| | Group for essential oils | 29 | 10.4 | 16 | 12.5 | 3 | 3.8 | 10 | 13.7 |
| | Group for others | 19 | 6.8 | 12 | 9.4 | 2 | 2.6 | 5 | 6.8 |

**Table A2.** Summary of NTFPs being cultivated, exploited, used, and traded in groups in the North Central, South Central, and Central Highland provinces (Unit: Species).

| Area | Province | Group for Yarn | Group for Food | Group for Medicinal Use | Group for Oil and Plastic | Group for Essential Oils | Group for Others |
|------|----------|----------------|----------------|-------------------------|---------------------------|--------------------------|------------------|
| North Central | Thanh Hoa (55) | 10 | 9 | 18 | 6 | 9 | 3 |
| | Nghe An (57) | 10 | 11 | 21 | 5 | 7 | 3 |
| | Ha Tinh (46) | 9 | 9 | 17 | 7 | 1 | 3 |
| | Quang Binh (30) | 8 | 5 | 8 | 4 | 2 | 3 |
| | Quang Tri (45) | 8 | 5 | 19 | 6 | 3 | 4 |
| | Thua Thien Hue (40) | 5 | 3 | 19 | 7 | 2 | 4 |
| | Total | 15 | 13 | 40 | 10 | 10 | 4 |
| South Central | Da Nang (32) | 10 | 5 | 7 | 2 | 5 | 3 |
| | Quang Nam (61) | 9 | 9 | 30 | 7 | 2 | 4 |
| | Quang Ngai (49) | 9 | 8 | 18 | 5 | 5 | 4 |
| | Binh Dinh (49) | 10 | 5 | 22 | 4 | 4 | 4 |
| | Phu Yen (26) | 4 | 4 | 13 | 2 | 1 | 2 |
| | Khanh Hoa (39) | 7 | 8 | 14 | 3 | 3 | 4 |
| | Ninh Thuan (34) | 6 | 7 | 13 | 2 | 4 | 2 |
| | Binh Thuan (22) | 5 | 6 | 6 | 2 | 1 | 2 |
| | Total | 14 | 16 | 43 | 9 | 10 | 4 |
| Central Highlands | Kon Tum (49) | 7 | 10 | 23 | 3 | 3 | 3 |
| | Gia Lai (51) | 8 | 12 | 21 | 3 | 4 | 3 |
| | Dak Lak (41) | 7 | 7 | 16 | 4 | 3 | 4 |
| | Dak Nong (22) | 5 | 4 | 6 | 2 | 3 | 2 |
| | Lam Dong (52) | 10 | 10 | 23 | 3 | 3 | 3 |
| | Total | 11 | 14 | 35 | 5 | 4 | 4 |

**Table A3.** Descriptive statistics of explanatory variables.

| Variable | Unit | All Households ($n = 400$) | | Planting Households ($n = 279$) | | Non-Planting Households ($n = 121$) | |
|----------|------|------|----------------------|------|----------------------|------|----------------------|
| | | Mean | Standard Deviation | Mean | Standard Deviation | Mean | Standard Deviation |
| PLAN | Yes = 1 | 0.69 | 0.45 | 1.00 | 0.00 | 0.00 | 0.00 |
| INCO$_{NTFPs}$ | Thousand dong | 9.36 | 10.93 | 13.43 | 10.80 | 0.00 | 0.00 |
| PERC | % | 10.22 | 11.78 | 14.65 | 11.57 | 0.00 | 0.00 |
| AGE | Years | 44.64 | 11.30 | 44.97 | 11.13 | 43.88 | 11.68 |
| EDUC | Years | 1.79 | 0.55 | 1.77 | 0.55 | 1.81 | 0.56 |
| GEND | Person | 0.41 | 0.52 | 0.40 | 0.53 | 0.44 | 0.49 |
| EXPE | Years | 8.24 | 4.94 | 9.18 | 5.16 | 6.08 | 3.54 |
| LABO | Person | 3.09 | 0.94 | 3.30 | 0.95 | 2.59 | 0.71 |
| FEMA | Person | 1.84 | 0.55 | 1.84 | 0.55 | 1.82 | 0.54 |
| EARN | % | 0.16 | 0.21 | 0.10 | 0.18 | 0.30 | 0.22 |
| CAPI | Thousand dong | 34.71 | 20.59 | 34.57 | 20.11 | 35.02 | 21.72 |
| BANK | Yes = 1 | 0.29 | 0.45 | 0.34 | 0.47 | 0.18 | 0.38 |
| INCO$_{AGRI}$ | Thousand dong | 53.43 | 46.19 | 57.37 | 49.61 | 44.34 | 35.69 |
| INCO$_{TIMBER}$ | Thousand dong | 31.74 | 34.92 | 32.15 | 33.96 | 30.81 | 37.19 |
| INCO$_{WAGE}$ | Thousand dong | 7.87 | 15.39 | 6.35 | 15.78 | 11.39 | 13.87 |
| INCO$_{OTHER}$ | Thousand dong | 0.41 | 0.85 | 0.39 | 0.90 | 0.44 | 0.70 |
| AREA | e | 5.21 | 4.28 | 5.20 | 4.28 | 5.22 | 4.31 |
| HOME | km | 5.85 | 2.48 | 5.36 | 2.42 | 6.98 | 2.25 |
| ROAD | km | 5.00 | 1.97 | 5.09 | 1.95 | 4.79 | 2.01 |
| SLOP | Degrees | 2.93 | 0.54 | 2.95 | 0.52 | 2.87 | 0.57 |
| SOIL | Very fair = 1 | 2.16 | 1.06 | 2.32 | 1.09 | 1.80 | 0.90 |
| INFO | Yes = 1 | 0.47 | 0.49 | 0.53 | 0.50 | 0.33 | 0.47 |
| TRAI | Yes = 1 | 0.45 | 0.49 | 0.50 | 0.50 | 0.32 | 0.46 |
| INFL | Very weak = 1 | 2.71 | 1.00 | 2.73 | 1.03 | 2.68 | 0.94 |
| REGI1 | | 0.40 | 0.49 | 0.45 | 0.49 | 0.26 | 0.44 |
| REGI2 | | 0.30 | 0.45 | 0.28 | 0.44 | 0.34 | 0.47 |
| NTFP1 | | 0.20 | 0.40 | 0.28 | 0.45 | 0.00 | 0.00 |
| NTFP2 | | 0.13 | 0.33 | 0.18 | 0.39 | 0.00 | 0.00 |
| NTFP3 | | 0.19 | 0.39 | 0.28 | 0.44 | 0.00 | 0.00 |
| NTFP4 | | 0.14 | 0.35 | 0.20 | 0.40 | 0.00 | 0.00 |
| NTFP5 | | 0.07 | 0.25 | 0.10 | 0.30 | 0.00 | 0.00 |
| NTFP6 | | 0.04 | 0.21 | 0.06 | 0.25 | 0.00 | 0.00 |

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
