# Peer review of "Determinants of Non-Timber Forest Product Planting, Development, and Trading: Case Study in Central Vietnam"

_forests, doi:10.3390/f11010116_

Round 1

Reviewer 1 Report

The article presents important issues for multifunctional forestry. As the title suggests, the Authors consider the region of Asia. It is a pity that in the discussion part the authors did not refer to the results of the work of researchers from other regions of the world. Such a solution would make the article interesting and have a slightly broader, more global context. In the "Introduction" section - line 40 - please check the correctness "400-500 USD / year" Is it possible? In the second part „Methods and data”- point 2.1. „Research objects” - could the Authors explain what they mean by "households around the forest area?" This factor is important for research searches, but is not defined. In what radius from the forest area were households sought? What intervals between forest and home were used for statistical analysis? 2.2. "Survey method" - The Authors state (line 108-109) that they conducted research in each village. In how many villages exactly? Later in the manuscript, information can be found that these interviews were conducted in 20 communes (line 112), so how many villages? Were they all villages in the studied radius or were they selected at random? 2.4. „Index selection” - lines 179-181 are a repeat from point 2.2. (lines 114-116). The results are described correctly, all tables I consider as an important, needed work element.

Author Response

Please find the attached response report

Reviewer 2 Report

The issue addressed in the paper discusses important and utilitarian research.  Research topics - determinants of non-timber forest product planting, development, and trading in central Vietnam - are consistent with the journal profile.

Such studies are partially analysed in literature. It would be worth presenting the state of the art in a broader way. I suggest a more dilligent  description of the research methods (concise, coherent research scenario).

I recommend:
- to compare different approaches in research on the non-timber forest production (NTPF) development; justify why the approach described in the paper was used;
- to explain and substantively justify the selection of features (variables) for analysis (and moreover, was a normal distribution obtained?, were the variables analyzed statistically /the tables contain only basic descriptive statistics/, outliers eliminated?);
- to explain if criteria for the representativeness of the research sample have been indicated (quantitative and qualitative approach?);
- to explain whether the correlation of the analyzed features has been verified and possibly excluded; 
- to explain in detail the consequences of regression model analysis;
- to explain whether verification by means of the ML - Binary Logit method and ML - Censored Normal (TOBIT) was performed;
- briefly explain whether there is a need to use, for instance, other methods.

that is, supplement the descriptive analysis (to compare the quantitative and qualitative approach, and finally, substantive justification).

The language of this paper is highly correct, however some descriptions would benefit from being more concise.

I also suggest that recommendations for specific, economic and practical, not only general (and not entirely clear) applications of this research should be provided.

Author Response

(The authors gave the same response as above.)
